# Model-Based Safety Analysis for the Fly-by-Wire System by Using Monte Carlo Simulation

**Zhong Lu [1],\*, Lu Zhuang [1], Li Dong [1] and Xihui Liang [2]**

[1]   College of Civil Aviation, Nanjing University of Aeronautics and Astronautics, Nanjing 211106, China;
   zhuanglu@nuaa.edu.cn (L.Z.); dong_li@nuaa.edu.cn (L.D.)
[2]   Department of Mechanical Engineering, University of Manitoba, Winnipeg, MB R3T5V6, Canada;
   Xihui.Liang@umanitoba.ca
\*   Correspondence: luzhong@nuaa.edu.cn

**Abstract:** Safety analysis is one of the important means to show compliance with airworthiness requirements. The traditional safety analysis methods are significantly dependent on analysts' skills and experiences. A model-based safety analysis approach is proposed for typical fly-by-wire (FBW) systems based on the system development model built via Simulink, by which the response of system performances can be simulated. The safety requirements of the FBW system are defined by presenting the thresholds of system performance metrics, and the effects of failure conditions on aircraft safety are determined according to the system response simulation by injecting failures or failure combinations into the Simulink model. The Monte Carlo simulation method is used to calculate the probability of unsafe conditions, whose effects are determined by the system response simulation with fault injections. Finally, a case study is used to illustrate the effectiveness and advantages of our proposed approach.

**Keywords:** system safety assessment; fly-by-wire system; fault injection; Monte Carlo simulation; dynamic behavior mode

---

## 1. Introduction

Safety is the most important characteristic of aviation products. The flight control system is a typical safety-critical system of modern aircraft, whose failures or malfunctions will lead to an unsafe flight path or structural failure preventing continued safe flight and landing. In the modern transport category of airplanes, fly-by-wire (FBW) systems have been widely used to replace hydro-mechanical ones. By utilizing the FBW system, pilots' commands are converted to electronic signals transmitted by wires to flight control computers, and control commands are calculated by flight control computers based on control laws to determine the movements of the actuators at each control surface. Therefore, the mechanical circuit consisting of rods, cables and pulleys is not required anymore, and the weight of the airplane can be reduced.

In aircraft or system development, the safety assessment process is an integral process that is used to show compliance with airworthiness requirements such as 14CFR/CS 23.1309, 14CFR/CS 23.1309, 14CFR 33.75, CS-E 510 and so on. At present, the safety assessment for civil airborne systems and equipment is usually conducted according to the standard ARP4761 issued by the Society of Automotive Engineer (SAE) [1,2]. In this document, it is recommended that traditional safety analysis techniques including Dependence Diagram Analysis (DDA), Fault Tree Analysis (FTA), Markov Analysis (MA), Failure Mode and Effect Analysis (FMEA) are applied in the safety assessment process [2]. These techniques are based on information synthesized from several sources including informal design models and requirements documents, and they are usually performed manually by

safety engineers whose experiences and skills will affect the analysis significantly. Therefore, the results of traditional safety analysis techniques are incomplete, inconsistent and highly subjective [3].

To overcome the deficiencies of the traditional techniques, model-based safety analysis (MBSA) has been proposed. MBSA focuses on modeling the system in a formal specification (model), which can be extended by injecting failure modes of the physical system or conducting safety analysis automatically. In this way, the completeness, consistency and correctness of the safety analysis results can be ensured, and the dependency on the engineers' skills and experiences can be avoided [4].

Over the past decade, different kinds of modeling tools have been applied in the modeling of the formal specification for MBSA. These tools can be classified into three categories [5], which are graphical modeling tools, system modeling languages and failure logic modeling techniques. Graphical modeling tools include Matlab-Simulink [6–8], Modelica [9,10], Petri Net [11–13] and SCADE [3,14]; system modeling languages include SysML [15,16], AADL [17–19], AltaRica [20–22], and NuSMV [23,24]; and failure logic modeling techniques include HiP-HOPS [25,26] and failure propagation approaches [27].

Although all the above-mentioned tools can be applied in the modeling of a formal specification, some of them are used to build models for system development, such as Matlab-Simulink, Modelica, and AADL; others are used to build models for failure analysis specifically, such as AltaRica, HiP-HOPS and FPTN. In the case that the models for system development are applied, the fault can be injected to the model directly and the consistency between system development and safety analysis can be maximized. The model of a flight control system used in development is usually expressed in system dynamics (control laws), which are given in the form of differential equations, transfer functions or state equations, and Matlab-Simulink is the most widely used tool in the development of control systems to build system dynamics models. Meanwhile, other MBSA tools such as AADL, AltaRica and HiP-HOPS are suitable for describing the failure propagations of avionics systems.

In this study, an MBSA method is proposed for typical FBW systems based on the system development model built via Simulink, by which the response of system performances can be simulated. The safety requirements of the FBW system are defined by presenting the thresholds of system performance metrics, and the effects of failure conditions on aircraft safety are determined according to the system response simulation by injecting failures or failure combinations in the Simulink model. The Monte Carlo simulation method is used to calculate the probability of failure conditions (unsafe conditions), whose effects are determined by the system response simulation with fault injections.

The system safety process of an aircraft is usually composed of four parts, which are Functional Hazard Assessment (FHA), Preliminarily Aircraft/System Safety Assessment (PASA/PSSA), Aircraft/System Safety Assessment (ASA/SSA) and Common Cause Analysis (CCA). This study focuses on the ASA/SSA process, the response of the Simulink model with fault injection is used to determine the failure effects of failure modes and their combinations from FMEA, and the Monte Carlo simulation method is applied to calculate the probability of top failure conditions instead of FTA, DDA and MA. The rest of this paper is structured as follows. In Section 2, the nominal (failure-free) model of the FBW system is presented by using Simulink, and the safety requirements are defined by giving the thresholds of system responses for performance metrics. In Section 3, the typical failure modes of the FBW system are modeled by Simulink, and the fault injecting approach is proposed to extend the nominal model. In Section 4, the probability calculation of unsafe conditions based on the Monte Carlo simulation is presented as a step-by-step procedure. In Section 5, a lateral-directional flight control system is used as a case study to show the accuracy and advantages of our proposed Monte Carlo simulation-based method. In Section 6, concluding remarks are presented.

## 2. Nominal Model of a Typical FBW System

The model of a typical FBW system under normal operating conditions, which is called the nominal or failure-free model, is built with Simulink in this section. Then, the system performance

responses under the failed configurations can be simulated by injecting the corresponding failure mode into the nominal model.

### 2.1. System Modeling via Simulink

In aircraft design, the development model of the FBW system as well as the aircraft dynamics are expressed by mathematical models such as state-space functions, transfer functions and differential equations, which are usually modeled with Simulink. In this section, a lateral-directional flight control system [6] is applied as an example to illustrate how to build the nominal model with Matlab-Simulink (2017a, MathWorks, Natick, MA, USA, 2017).

The lateral-directional flight control system is composed of a flight control computer subsystem, an actuation subsystem, a sensor subsystem and control surfaces. The flight control computer subsystem has a dual redundant architecture that is composed of two identical primary flight computers (PFCs). The actuation subsystem includes the left aileron actuation (LAA), the right aileron actuation (RAA) and the rudder actuation (RA). Each of them is composed of two redundant actuators that are connected with a combiner. The sensor subsystem includes the right aileron position sensors (RAPS), the left aileron position sensors (LAPS), the rudder position sensors (RPS) and the inertial measurement units (IMU). All of them have a triple modular redundant architecture. The control surfaces for lateral control include the left aileron, the right aileron and the rudder. Pilots' commands from pedals and control sticks are converted to electronic signals transmitted by wires to the two PFCs, and control commands are calculated in terms of control laws in the PFCs to determine the movements of the actuators for all control surfaces. The architecture of the lateral-directional flight control system is shown in Figure 1.

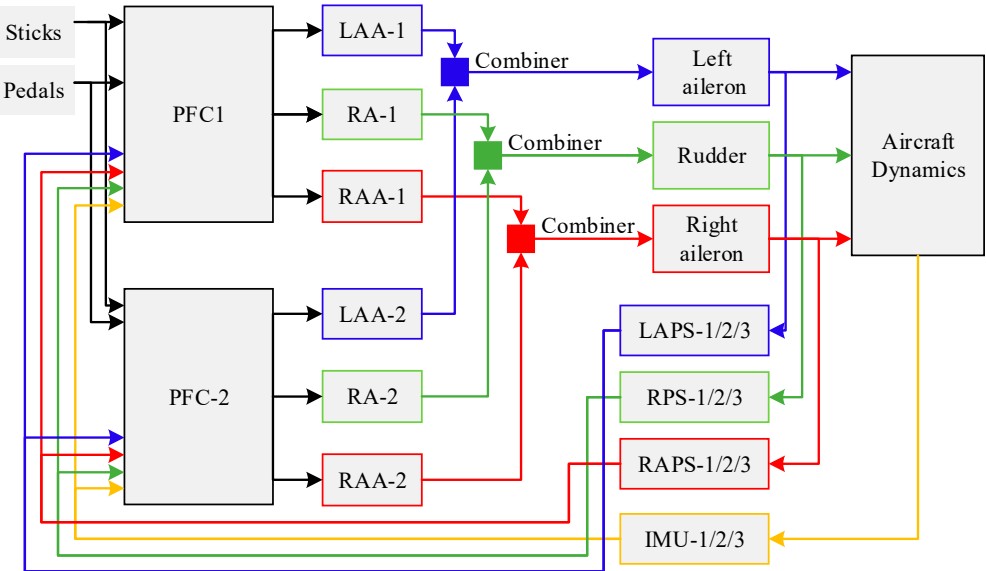

**Figure 1.** Architecture of the lateral-directional flight control system.

According to the dynamic models of the lateral-directional flight control system given in [6,28], the corresponding Simulink model can be built. Taking the roll control law of the PFC as an example, the mathematical model is expressed as

$$\begin{aligned}
R_r(s) &= K_{r_1}\phi_c(s) + K_{r_2}R_b(s) + K_{r_3}\frac{s+z_r}{s+p_r}P_b(s)\\
\delta_{a_r^*}^{l(r)}(s) &= \left(P_r + \frac{I_r}{s} + D_r s\right)\left(R_r(s) + K_r\delta_a^{l(r)}(s)\right)
\end{aligned} \tag{1}$$

where $\phi_c(\cdot)$ is the roll command, $R_b(\cdot)$ is the yaw rate, $P_b(\cdot)$ is the roll rate, $\delta_a^{l(r)}(\cdot)$ is the angle of the left (right) aileron, and $\delta_{a_r^*}^{l(r)}(\cdot)$ is the output response of the roll control law. The values of the coefficients

are given as $K_{r_1} = 0.66$, $K_{r_2} = -0.145$ s, $K_{r_3} = 2.16$ s, $z_r = 11.1$ s$^{-1}$, $p_r = 25$ s$^{-1}$, $P_r = 0.45$ A, $I_r = 6$ A/s, $D_r = 0.01$ As, and $K_r = -1.33$ [6,28].

The corresponding nominal Simulink model is shown in Figure 2.

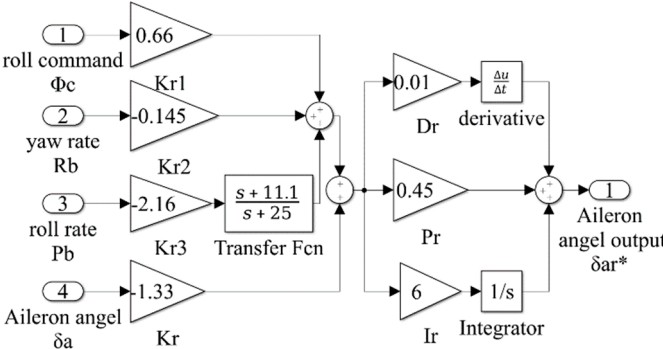

**Figure 2.** Simulink model of the roll control law of the primary flight computer (PFC).

By combining the Simulink models of all components, we can build the Simulink model of the lateral-directional flight control system, which is shown in Figure 3. In both Figures 2 and 3, the symbol "*" is used to note the output variables of the control laws.

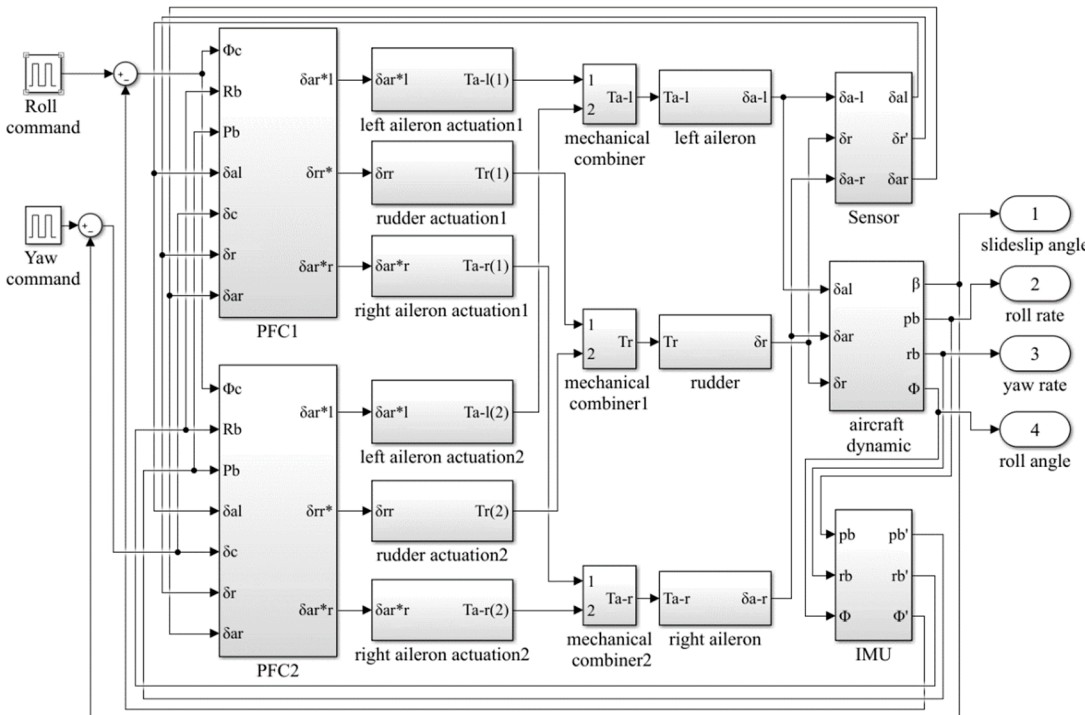

**Figure 3.** Nominal Simulink model of the lateral-directional flight control system.

### 2.2. The Definition of the FBW System Safety Requirement

When the aircraft is in a safe condition, the output response of each performance metric should be restricted to within an acceptable region, in which the requirements of the performance metrics can be satisfied. For the lateral-directional flight control system, the performance metrics include the sideslip

angle $\beta(t)$, the roll rate $p_b(t)$, the yaw rate $r_b(t)$ and the roll angle $\phi(t)$, thus the safety requirement of the system can be defined as

$$
\begin{cases}
\left| \beta(t) - \beta_r(t) \right| \leq r_\beta \\
\left| p_b(t) - p_{b_r}(t) \right| \leq r_{p_b} \\
\left| r_b(t) - r_{b_r}(t) \right| \leq r_{r_b} \\
\left| \phi(t) - \phi_r(t) \right| \leq r_\phi
\end{cases}
\tag{2}
$$

where $\beta_r(t)$, $p_{b_r}(t)$, $r_{b_r}(t)$ and $\phi_r(t)$ are the reference values of the sideslip angle, the roll rate, the yaw rate and the roll angle, respectively, which are the responses of these parameters in the failure-free configuration; $r_\beta$, $r_{p_b}$, $r_{r_b}$ and $r_\phi$ are the thresholds of the sideslip angle, the roll rate, the yaw rate and the roll angle, respectively. Here, we take $r_\beta = 0.15$ rad, $r_{p_b} = 0.45$ rad/s, $r_{r_b} = 0.45$ rad/s and $r_\phi = 0.15$ rad from [28].

## 3. Extension of the Nominal Model

The objective of extending the nominal model is to inject different kinds of failure modes into the failure-free model. In this way, the performance responses of the FBW system under failed configurations can be obtained, and the unsafe conditions can be determined by comparing these responses with the performance thresholds given in Equation (2). In this section, the failure modes as well as their mathematical model are given, the Simulink tool is also used to build the models of different kinds of failure modes, and the fault injecting method is proposed.

### 3.1. Failure Modes and Their Mathematical Model

The failure modes of the components of the FBW system, as well as their failure rates, are given in Table 1.

**Table 1.** Component failure modes of the fly-by-wire (FBW) system [6].

| Component | Failure Mode | Failure Mode Description | Failure Rate (1/h) |
|---|---|---|---|
| PFCs | Omission | Null output | $2 \times 10^{-7}$ |
| | Random | Random output between −5 and 5 | $1 \times 10^{-7}$ |
| | Stuck | Output stuck at the last correct value | $1 \times 10^{-7}$ |
| | Delayed | Output delayed by 0.2 s | $1 \times 10^{-7}$ |
| Actuators | Omission | Null output | $1 \times 10^{-6}$ |
| | Stuck | Output stuck at the last correct value | $1 \times 10^{-6}$ |
| Control Surfaces | Stuck | Output stuck at the last correct value | $1 \times 10^{-8}$ |
| | Trailing | Output decided by the aero-dynamics | $1 \times 10^{-8}$ |
| Inertial Measurement Units (IMU) | Omission | Null output | $4 \times 10^{-7}$ |
| | Gain change | Output scaled by a factor of 1.5 | $3 \times 10^{-7}$ |
| | Biased | Output biased by a factor of 0.3 deg | $3 \times 10^{-7}$ |
| Position Sensors | Omission | Null output | $4 \times 10^{-7}$ |
| | Gain change | Output scaled by a factor of 1.5 | $3 \times 10^{-7}$ |
| | Biased | Output biased by a factor of 0.3 deg | $3 \times 10^{-7}$ |

Reproduced with permission from (Dominguez-Garcia A. D., Kassakian J. G., Schindall J. E., et al.), (Reliability Engineering & System Safety); published by (Elsevier), 2008.

The dynamic behavior of each component can be expressed as the state-space function:

$$
\begin{cases}
\dot{\boldsymbol{x}}(t) = \boldsymbol{Ax}(t) + \boldsymbol{Bu}(t) \\
\boldsymbol{y}(t) = \boldsymbol{Cx}(t) + \boldsymbol{Du}(t)
\end{cases}
\tag{3}
$$

where $\boldsymbol{x}(t)$ is the vector of state variables, $\boldsymbol{u}(t)$ is the vector of input variables, $\boldsymbol{y}(t)$ is the vector of output variables, $A$ is the system matrix, $B$ is the control matrix, $C$ is the output matrix and $D$ is the feedforward matrix.

We make the assumption that the output of the failure-free configuration is $y(t)$, and the output of different failure modes can be expressed as

$$\bar{y}(t) = \begin{cases} 0 & \text{ommision} \\ rand & \text{random} \\ y(\tau_s) & \text{stuck} \\ y(t - \tau_d)1(t - \tau_d) & \text{delayed} \\ \vartheta \text{ or } \psi & \text{trailing} \\ Gy(t) & \text{gain change} \\ y(t) + b & \text{biased} \end{cases} \tag{4}$$

where rand is a random value expressing the random output, $\tau_s$ is the time point when "stuck" occurs, $\tau_d$ is the delayed time, $\vartheta$ is the pitching angle (for the ailerons), $\psi$ is the heading angle (for the rudder), $G$ is the gain change factor and $b$ is the biased factor.

## 3.2. Failure Mode Modeling via Simulink

The Simulink tool is also used to build the models of the seven failure modes expressed via Equation (4), which are given in Figure 4.

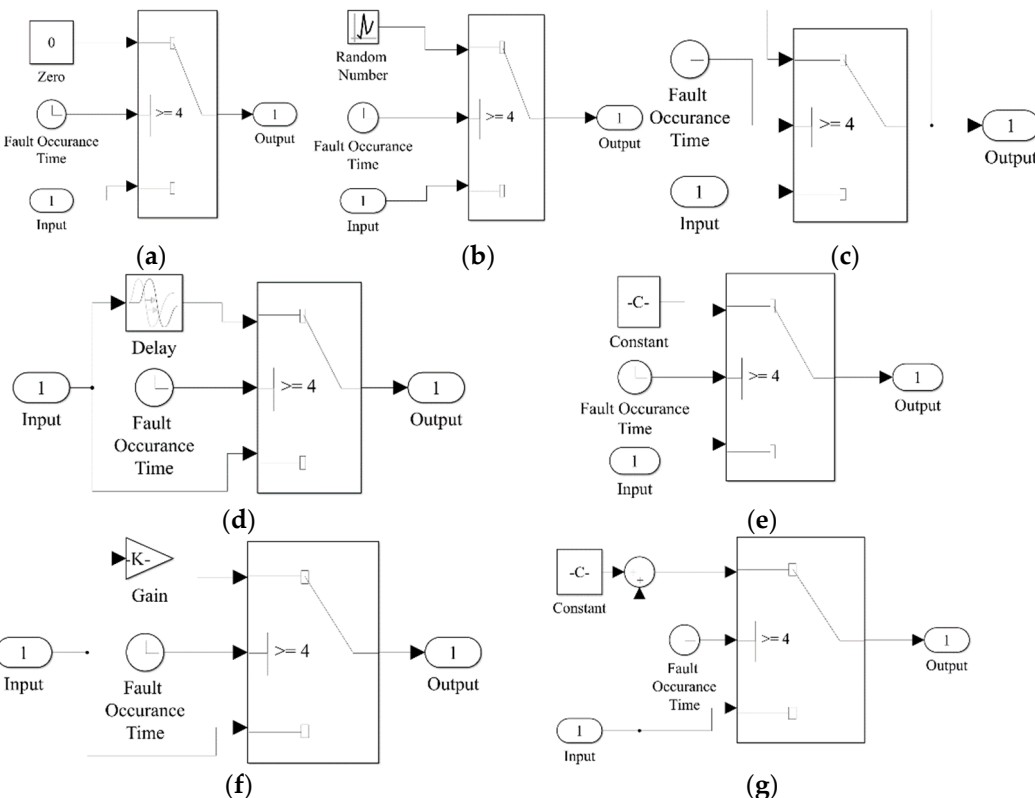

**Figure 4.** Simulink models of the seven failure modes: (**a**) "omission", (**b**) "random", (**c**) "stuck", (**d**) "delayed", (**e**) "trailing", (**f**) "gain change", and (**g**) "biased".

In all the models, the time point of fault injection is set as the 4th second, and the input is the failure-free response of the related component. Therefore, the output of each model before the 4th second will be the failure-free response of the related component as well. Figure 4a shows the Simulink model of "omission"; the "Output" block will connect to the "Zero" block 4 s later, thus the output after the 4th second will be null. Figure 4b shows the Simulink model of "random"; the "Output" block will connect to the "Random Number" block 4 s later, then the output after the 4th second will be a random value. Figure 4c shows the Simulink model of "stuck"; the "Output" block will connect to the

output value of "4 s" after the 4th second, then the output will be stuck at its value of "4 s". Figure 4d shows the Simulink model of "delay"; the "Output" block will connect to the "Delay" block 4 s later, then the output after the 4th second will be delayed. Figure 4e shows the Simulink model of "trailing"; the "Output" block will connect to the "Constant" block 4 s later, then the output after the 4th second will be a constant value. The constant value C in the"Constant" block will be the heading angle $\psi$ for the rudder and the pitching angle $\vartheta$ for the ailerons. Figure 4f shows the Simulink model of "gain change"; the "Output" block will connect to the "Gain" block 4 s later, then the output will be scaled by the gain change factor, which is shown as K in the "Gain" block. Figure 4g shows the Simulink model of "biased"; the "Output" block will connect to the "Sum" block 4 s later, then the output will be the summation of the initial output and the biased factor, which is shown as C in the "Constant" block.

### 3.3. The Fault Injecting Method

The fault injection is conducted by a "fault injector", which is a "Variant Subsystem" block placed between each component block and its output. Figure 5 shows the fault injector and its inner structure. The fault injector includes the blocks of the seven failure modes given in Figure 4a–g as well as an additional "normal" block. The "normal" block denotes the failure-free configuration; its inner "Input" connects "Output" directly.

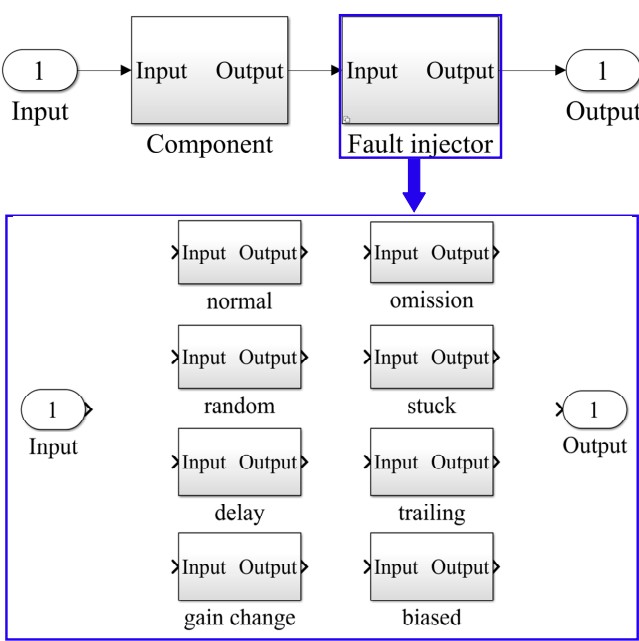

**Figure 5.** Simulink model of the fault injector.

The block parameters of the "fault injector" are used as the control variables for fault injections. By selecting different control variables, the responses of different kinds of configurations can be obtained.

Figure 6 shows the performance responses of the FBW system. Figure 6a shows the performance responses of the FBW system in the failure-free configuration when the roll command is a 0.2 rad, 0.1 Hz square wave. Figure 6b shows the system performance responses of the same input command when the "random" failure mode of one PFC has been injected. We can see that the difference between the roll angle response and its reference value has exceeded the threshold (0.15 rad). Thus, there is an unsafe condition. The results shown in these figures are similar to those in [6]. Figures 6a and 6b here correspond to Figures 2b and 6b in [6], respectively.

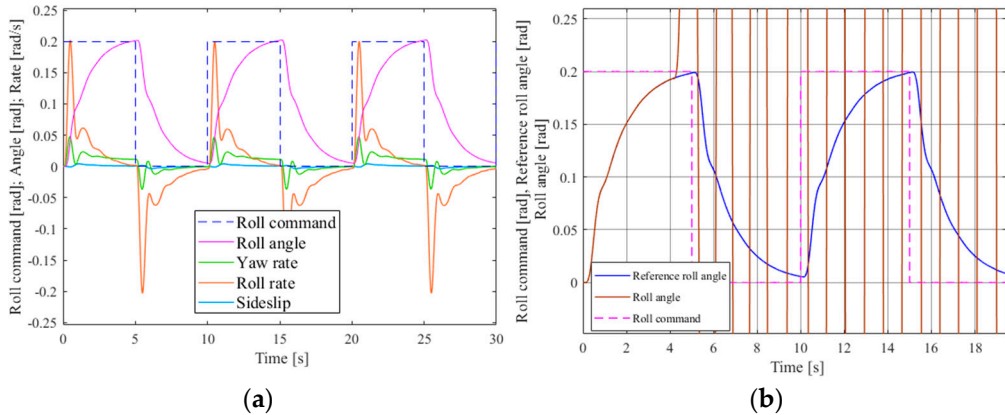

**Figure 6.** Performance responses of the FBW system: (**a**) the failure-free configuration; (**b**) the configuration of the "PFC random" failure mode.

## 4. Probability Calculation of Unsafe Conditions Based on Monte Carlo Simulation

Random numbers are used to denote the failure time for each component failure mode. These numbers will be the fault occurrence time of each corresponding failure mode. We order these numbers from smallest to largest, and inject the failure modes at their occurrence time one by one according to the fault injecting method. When the responses do not satisfy the safety requirement given in Equation (2), one simulation will terminate and a sample of time to the unsafe condition can be obtained. In terms of several time samples, the probability distributions of the time to unsafe conditions can be obtained. In this way, the probability of unsafe conditions can be calculated at different time points.

We make the assumption that the system is composed of n components and the *i*th ($i = 1, 2, \ldots , n$) component has $m_i$ failure modes. $N$ is used to denote the ordinal of the current simulation. $T_N$ is the time to unsafe conditions obtained from the $N$th simulation.

The step-by-step procedure of the Monte Carlo simulation-based method is as follows. The flowchart of the step-by-step procedure is shown in Figure 7.

Step 1    Initialization

Before the simulation starts, the ordinal of the current simulation is zero, namely $N = 0$; the corresponding time to unsafe conditions is also set as zero, namely $T_0 = 0$.

Step 2    Start a new simulation

Let $N = N + 1$, and the $(N + 1)$th simulation will start.

Step 3    Generate random numbers

Generate random numbers as the failure time for all failure modes. A common situation is that the failure time of each failure mode follows the exponential distribution, and the random number can be expressed as

$$t_{ij} = -\frac{1}{\lambda_{ij}} \ln(1 - r_{01}) \ (i = 1, 2, \cdots , n; \ j = 1, 2, \cdots , m_i) \tag{5}$$

where $\lambda_{ij}$ is the constant failure rate of the *i*th component's *j*th failure mode, $r_{01}$ is a random number generated from the uniform distribution defined over the interval (0, 1), and $t_{ij}$ is the failure time of the *i*th component's *j*th failure mode. For other distributions, we can also obtain a random number by their cumulative density function.

Step 4    Obtain the failure time for each component

As each component has several failure modes and the component will fail when one of the failure modes occurs, the failure time of each component will be the minimum value of all its failure modes' occurrence time. We have

$$t_{ij_i} = \min_{j=1,2,\cdots,m_i} \left( t_{ij} \right) \tag{6}$$

where $t_{ij_i}$ is the failure time of the $i$th component, and $j_i$ is the ordinal of the $i$th component's failure mode that has the minimum failure time.

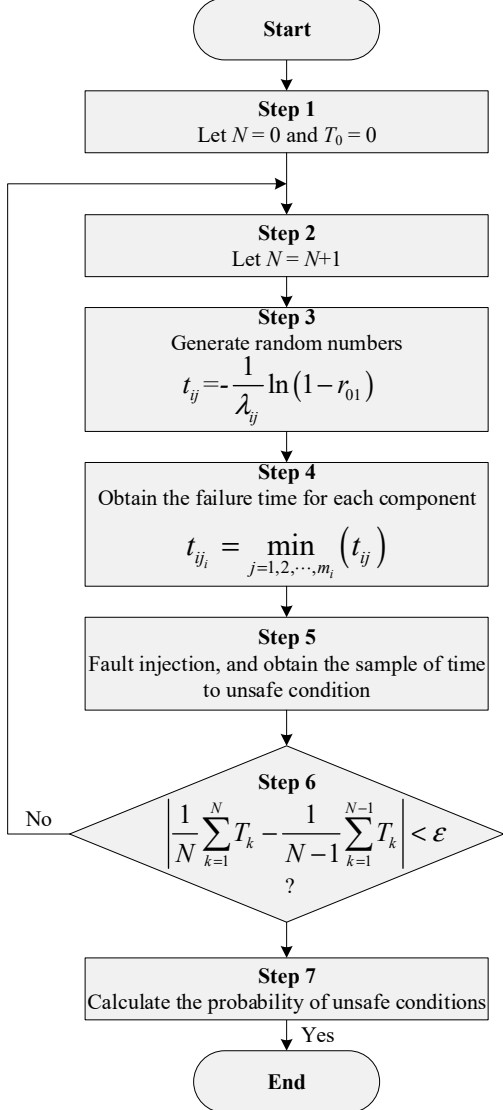

**Figure 7.** Flowchart of the Monte Carlo simulation.

Step 5　Inject the fault of each component

The failure time for each component is ordered from smallest to largest, and the $i$th component's $j_i$th failure mode will be injected into the nominal model at $t_{ij_i}$ one by one according to the fault injection method proposed in Section 3. When the unsafe condition occurs, the corresponding $t_{ij_i}$ will be the $N$th sample of the time to unsafe conditions obtained from the $N$th simulations ($T_N$).

Step 6　Decide whether the simulations will end or not

The simulations will stop when the mean time to unsafe conditions converges, thus the simulation ending criteria can be expressed as

$$\left| \frac{1}{N} \sum_{k=1}^{N} T_k - \frac{1}{N-1} \sum_{k=1}^{N-1} T_k \right| < \varepsilon \tag{7}$$

where $\varepsilon$ is an arbitrarily small positive real, and we usually let $\varepsilon = 0.1$.

If Equation (7) can be satisfied, the simulation procedure will go to Step 7; otherwise, it will go back to Step 2.

Step 7    Calculate the probability of unsafe conditions

According to the $N$ samples of time to unsafe conditions, we perform distribution selections, parameter estimations and goodness-of-fit tests. Then, we can obtain the probability distribution of time to unsafe conditions, and the probability of the time to unsafe conditions can be calculated.

## 5. Case Study and Discussion

The lateral-directional flight control system given in Figure 1 is applied as the case study here. The failure rates of all components' failure modes are already given in Table 1.

By conducting the Monte Carlo simulation procedure given in Section 4, we obtain nearly 2000 samples of time to unsafe conditions, whose histogram is shown in Figure 8. According to the shape of the histogram, it can be estimated that these samples may follow Weibull distribution or Lognormal distribution. For the $k$th sample $T_k$, the estimate of the related cumulative distribution function (CDF) can be expressed as

$$\hat{F}(T_k) = \frac{k}{N} \tag{8}$$

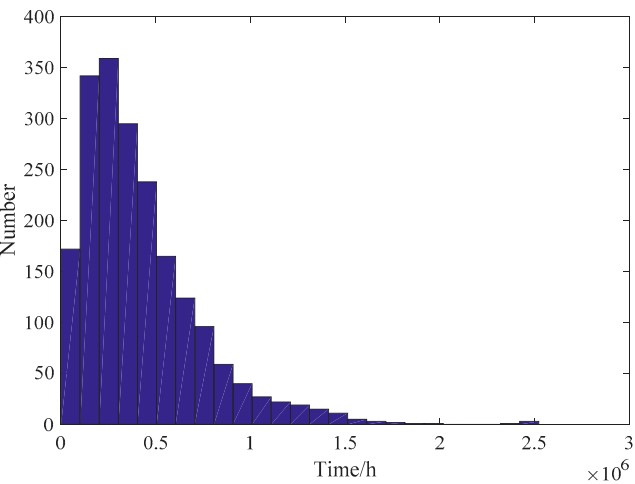

**Figure 8.** Histogram of time samples.

The CDF of Weibull distribution is expressed as

$$F(t) = 1 - \exp\left[-\left(\frac{t}{\alpha}\right)^{\beta}\right] \tag{9}$$

We let

$$\begin{cases} x_k = \ln T_k \\ y_k = \ln\left\{\ln\left[\frac{1}{1-\hat{F}(T_k)}\right]\right\} \end{cases} \tag{10}$$

Thus, if the samples follow a Weibull distribution, the plotting of $x_k$ versus $y_k$ expressed by (10) should look like a straight line. This plotting is shown in Figure 9a.

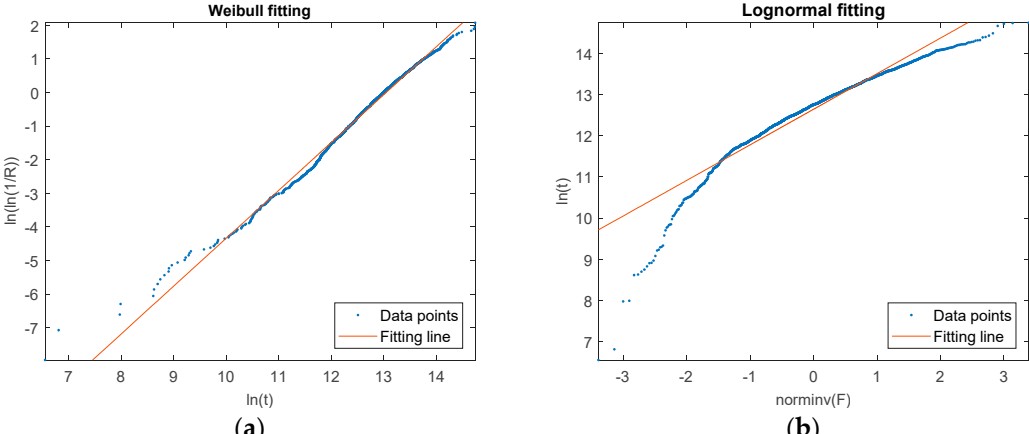

**Figure 9.** (**a**) The plotting of Weibull fitting; (**b**) The plotting of lognormal fitting.

Likewise, the CDF of a lognormal distribution is expressed as

$$F(t) = \Phi\left(\frac{\ln t - \mu}{\sigma}\right) \tag{11}$$

where $\Phi(\cdot)$ is the CDF of the standard normal distribution. Let

$$\begin{cases} x_k = \Phi^{-1}\left[\hat{F}(T_k)\right] \\ y_k = \ln T_k \end{cases} \tag{12}$$

Thus, if the samples follow a lognormal distribution, the plotting of $x_k$ versus $y_k$ expressed by (12) should look like a straight line. This plotting is shown in Figure 9b.

It is shown that the plotting of $x_k$ versus $y_k$ in Figure 9a is much more like a straight line compared with Figure 9b. Therefore, the Weibull distribution is preferred for these samples.

By using the maximum likelihood estimation, we can obtain $\alpha = 2.6062 \times 10^5$ h and $\beta = 1.211$. In addition, the Kolmogorov–Smirnov test shows that we should not reject the hypothesis that the samples of time to unsafe conditions are following the Weibull distribution with $\alpha = 2.6062 \times 10^5$ and $\beta = 1.211$ [29]. Hence, the probability of the unsafe condition at time t can be expressed as

$$P(t) = 1 - \exp\left[-\left(\frac{t}{2.6062 \times 10^5}\right)^{1.211}\right] \tag{13}$$

For the FBW system, the scheduled inspection interval is set as 500 flight hours, which means the function of the FBW system will be thoroughly checked every 500 flight hours and the FBW system will be restored to the perfect condition if it is degraded. Thus, we can obtain the probability of the unsafe condition in a scheduled inspection interval as $5.1234 \times 10^{-4}$. Therefore, the average probability of the unsafe condition per flight hour is $1.0247 \times 10^{-6}$.

In [6], the Markov process is applied to calculate both the upper and lower bounds of the probability of the unsafe condition in the scheduled inspection interval, which are $5.1178 \times 10^{-4}$ and $5.8211 \times 10^{-4}$, respectively, for this case. We have also used the state enumerating method to obtain the minimal cut sets that cause unsafe conditions. In the state enumerating method, we simulate the model of the FBW system with all component failure modes and their combinations, and the minimal cut sets can be obtained in terms of the system response. Moreover, the probability of the unsafe condition or its interval can be calculated via the probability additive formula. The upper and lower

bounds of the probability of the unsafe condition calculated by the minimal cut sets are $5.1088 \times 10^{-4}$ and $5.2084 \times 10^{-4}$, respectively. The results of the three methods are given in Table 2.

**Table 2.** Probability of the unsafe condition calculated by different methods.

| Monte Carlo Simulation | Markov Process | State Enumerating |
|:---:|:---:|:---:|
| $5.1234 \times 10^{-4}$ | $(5.1178 \times 10^{-4}, 5.8211 \times 10^{-4})$ | $(5.1088 \times 10^{-4}, 5.2084 \times 10^{-4})$ |

Table 2 shows that the result of the Monte Carlo simulation method is located just between the upper and lower limits obtained from both the Markov process and the state enumerating, which illustrates the accuracy of our Monte Carlo simulation method. The advantages and disadvantages of the above-mentioned three methods are discussed in Table 3.

**Table 3.** Comparison of the three methods.

| Methods | Advantages | Disadvantages |
|:---:|:---|:---|
| Monte Carlo simulation | • The cumbersome work of building a Markov model can be avoided.<br>• The simulation algorithm does not have to be modified when the system is modified. | • Time-consuming for large systems.<br>• A stochastic method, and it does not achieve exact results. |
| Markov process | • The exact value or interval of the probability can be calculated.<br>• More efficient than Monte Carlo simulation after the Markov model has been built. | • Faced with the state explosion problem.<br>• A new Markov model is needed when the system is modified. |
| State enumerating | • The probability additive formula is used to calculate the exact value or interval of the probability for an unsafe condition. | • Impossible to enumerate all the states for large systems.<br>• A new formula is required when the system is modified. |

## 6. Conclusions

In this study, an MBSA approach is proposed based on the system development model built by Simulink for the FBW system, and Monte Carlo simulation is used to obtain the probability of unsafe conditions. Our proposed approach has the following advantages:

(1) The performance responses of the system with fault injection are used to determine the effect of component failures or failure combinations on system safety. Compared with the traditional safety analysis methods, the determination of failure effects is no longer dependent on analysts' specific knowledge about the aircraft system.

(2) By using the Monte Carlo simulation method, the cumbersome work of building a Markov model can be avoided, and the state explosion problem of the Markov process can be resolved to some extent. Additionally, when the system is modified or changed, the Markov model should be rebuilt; however, our Monte Carlo simulation algorithm should not be updated.

(3) By using the system development model built by Simulink, the safety assessment can be carried out in the early stage of system development. Moreover, it is easy to update the safety assessment results with the design improvement of the FBW system.

**Author Contributions:** Conceptualization, Z.L.; methodology, Z.L. and L.Z.; software, Z.L. and X.L.; formal analysis, Z.L., L.Z. and X.L.; data curation, L.Z. and L.D.; writing—original draft preparation, Z.L. and L.D.; project administration, Z.L.; funding acquisition, Z.L. All authors have read and agreed to the published version of the manuscript.

**Funding:** This research was supported in part by the National Natural Science Foundation of China under Grant U1733124, in part by the Aeronautical Science Foundation of China under Grant 20180252002, and in part by the Fundamental Research Funds for the Central Universities of NUAA under Grant 3082018NT2018019.

**Conflicts of Interest:** The authors declare no conflict of interest.

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
