# Peer review of "Model-Based Safety Analysis for the Fly-by-Wire System by Using Monte Carlo Simulation"

_processes, doi:10.3390/pr8010090_

Round 1

Reviewer 1 Report

The proposed article, named "Model-based Safety Analysis for the Fly-by-Wire System by using Monte Carlo Simulation", presents a way to study the safety performances of an aircraft part: the "fly-by-wire" system. This proposed safety analysis is based on stochastic simulation of a Simulink model. The Simulink model represents the nominal operation of the system, extended with failure modes.
This article is clear, well organized and well written. The state of the art part of the introduction presents the most relevant technologies of the domain of MBSA. But the rest of the article does not compare the results of the proposed approach to existing methodologies: the AltaRica technology has a stochastic simulator, as well as the Petri net technology Griff, or the Hip-HOPS technology which is based on Matlab-Simulink.
But finally the most critical aspect of the explained approach is how the safety analysis of a system (a subpart of an aircraft here) is taken into account. In fact, the classical approach starts with the main functions of the considered system defined by the system design team. It first defines their non-nominal operating. This leads to the consequences of these non-nominal operations in terms of SIL levels (or DAL for Airborne Systems). These studies are classically provided by FMEA: the Failure Modes of a system and the Analysis of their Effects on the system itself and its environment. Then for critical failure modes, we study the causes of them: which basic events (from the component point of view) or combination of basic events lead to the failure modes. The proposed approach explained in this article only considers the FMEA part, not the study of the causes which is classically the most important part of a safety analysis. Thus it must be clarified which part of the safety analysis is considered with the proposed approach.

Author Response

Thanks for your comments. We think that the comments contain two questions. The first one is why other MBSA methods such as AltaRica, Hip-HOPs, Petri Net are not used here. And the second one is which part of the safety analysis is considered with the proposed approach.

To the first question, we have added the following explanations. (See the highlighted part on page 2).

“The model of a flight control system used in development is usually expressed system dynamics (control laws), which are given in the forms of differential equations, transfer functions or state equations, and Matlab-Simulink is the most widely used tools in the development of control systems to build the model of system dynamics. While other MBSA tools such as AADL, AltaRica and HiP-HOPS, are suitable for describing the failure propagations of avionics systems.”

To the second question, we have added the following explanations. (See the highlighted part on page 2).

The system safety process of the aircraft is usually composed of four parts, which are Functional Hazard Assessment (FHA), Preliminarily Aircraft/System Safety Assessment (PASA/PSSA), Aircraft/System Safety Assessment (ASA/SSA) and Common Cause Analysis (CCA). This study is focusing on the ASA/SSA process, the response of the Simulink model with fault injection is used to determine the failure effects of failure modes and their combinations from FMEA, and the Monte Carlo simulation method is applied to calculate the probability of top failure conditions instead of FTA, DDA and MA.

Reviewer 2 Report

The paper is generally well set-out but should emphasise more fully the originality of applying Monte Carlo simulation to the assessment of system safety.

How are the failure rates in Table 1 derived, what source is used? Line 124, explain what is meant by "acceptable region" Line 193, it would be good to quote from Ref [6] I.e. show a figure from there so the similarity can be seen. Line 268, what hypothesis is being assessed?  This is unclear. Line 270, what is meant by the Inspection interval?  The FBW system is very extensive, so does this mean functional check - explain. Line 275 / Table 2, explain or rephrase "State enumerating method" Section 6 - the conclusions should be expanded to more clearly explain the advantages and applications of the technique.  Line 291, rephrase "...not depending on analysts' experiences and skills anymore".  The new approach you explained DOES need skill and experience, but maybe less aircraft specific knowledge?

Some editing required

Figure 1 should be labelled "Left aileron"/"Right aileron", delete "The" Line 109 should be "Taking" not "take" Line 111 replace "angel" with "angle" and check rest of paper e.g. Figure 2, line 191 etc Line 113 delete "And" Line 145, "out" should be "output" Line 149, rephrase "when the stuck occurs" Line 206 should be "mi failure modes" Line 251, delete "we can roughly determine..." and replace with "it can estimated that..." Table 2, insert space after table. Table 3, add horizontal lines so that rows can be separated

Author Response

Question 1:

The paper is generally well set-out but should emphasize more fully the originality of applying Monte Carlo simulation to the assessment of system safety.

Response: Thanks for pointing out the problem. The statement of the originality has been added as “By using the Monte Carlo simulation method, the cumbersome work of building a Markov model can be avoided, and the state explosion problem of the Markov process can be resolved to some extent. Additionally, when the system is modified or changed, the Markov model should be rebuilt, however, our Monte Carlo simulation algorithm should not be updated.” (See the highlighted part on page 12)

Question 2:

How are the failure rates in Table 1 derived, what source is used?

Response: Table 1 is reprinted form reference [6]. We have got the permission of reuse this table from Elsevier. Additionally, A statement “Reproduced with permission from [Dominguez-Garcia A. D., Kassakian J. G., Schindall J. E., et al], [Reliability Engineering & System Safety]; published by [Elsevier], [2008].” has been added below the table.

Question 3:

Line 124, explain what is meant by "acceptable region"

Response: the “acceptable region” means the performance metrics requirements. We have added the explanation “in which the requirements of the performance metrics can be satisfied”. (See the highlighted part on page 4)

Question 4:

Line 193, it would be good to quote from Ref [6] I.e. show a figure from there so the similarity can be seen.

Response: We think our manuscript will be too trivial if we show the figures of Ref [6] here. In our manuscript, Fig.6 (a) and Fig.6 (b) are similar to Fig. 2 (b) and Fig. 6 (b) of Ref. [6], respectively. The explanation has been added. (See the highlighted part on page 7)

Question 5:

Line 268, what hypothesis is being assessed? This is unclear.

Response: the hypothesis is that the samples of time to unsafe conditions follows the Weibull distribution. The explanation has been added. (See the highlighted part on page 11)

Question 5:

Line 270, what is meant by the Inspection interval? The FBW system is very extensive, so does this mean functional check - explain.

Response: the inspection interval 500 flight hours is called A-check in aircraft maintenance, which means the function of the FBW will be thoroughly checked every 500 flight hours. If the FBW system is in a degraded condition, it will be restored to its initial condition by some maintenance tasks. Explanations has been added. (See the highlighted part on page 11)

Question 6:

Line 275 / Table 2, explain or rephrase "State enumerating method" Section 6 - the conclusions should be expanded to more clearly explain the advantages and applications of the technique.  

Response: In the state enumerating method, we simulate the FBW system with all component failure modes and their combinations, and the minimal cut sets can be obtained in terms of the system response. Moreover, the probability of the unsafe condition or its interval can be calculated via the probability additive formula. The explanations has been added. (See the highlighted part on page 11)

Question 7:

Line 291, rephrase "...not depending on analysts' experiences and skills anymore". The new approach you explained DOES need skill and experience, but maybe less aircraft specific knowledge?

Response: thanks for pointing out this. We have revised the statement as “the determination of failure effects is not depending on analysts’ specific knowledge about the aircraft system anymore.” (See the highlighted part on page 11)

Some editing required

Figure 1 should be labelled "Left aileron"/"Right aileron", delete "The"; Line 109 should be "Taking" not "take"; Line 111 replace "angel" with "angle" and check rest of paper, e.g. Figure 2, Line 191 etc. Line 113 delete "And"; Line 145, "out" should be "output"; Line 149, rephrase "when the stuck occurs"; Line 206 should be "mi failure modes"; Line 251, delete "we can roughly determine..." and replace with "it can estimated that..."; Table 2, insert space after table; Table 3, add horizontal lines so that rows can be separated.

Response:

All the above editing errors have been corrected, thanks for pointing all of them.

Round 2

Reviewer 1 Report

The 2nd version of the publication "Model-based Safety Analysis for the Fly-by-Wire System by using Monte Carlo Simulation" improves the 1st one by explaining more better the position of the proposed approach according to safety standard of aircraft.